# Quality-Preserving Auto-Regressive Mesh Generation Acceleration via Multi-Head Speculative Decoding

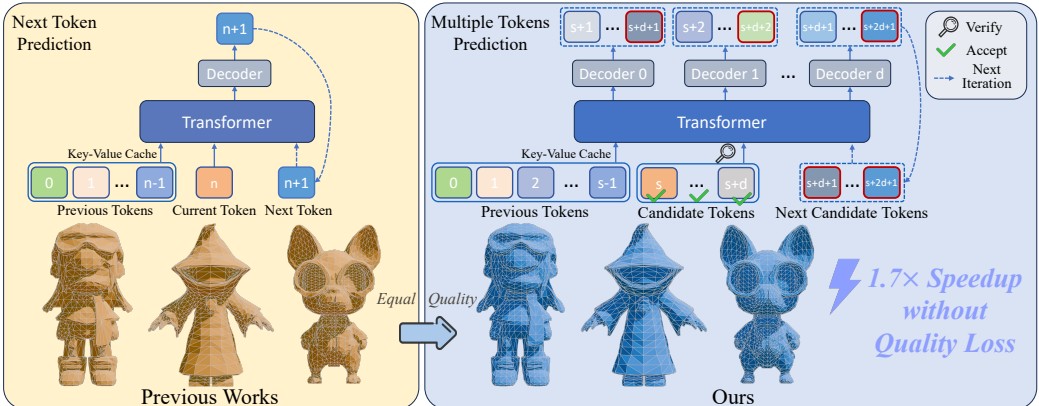

Figure 1: **The differences between our framework and previous works.** We propose XSpecMesh, a method for accelerating auto-regressive mesh generation models via multi-head speculative decoding, instead of relying on traditional next-token prediction. In a single forward pass, multiple decoding heads predict several tokens, verify the candidate tokens, and resample candidate tokens for the next iteration. Our approach delivers a $1.7\times$ speedup while preserving generation quality.

## Abstract

Current auto-regressive models can generate high-quality, topologically precise meshes; however, they necessitate thousands—or even tens of thousands—of next-token predictions during inference, resulting in substantial latency. We introduce XSpecMesh, a quality-preserving acceleration method for auto-regressive mesh generation models. XSpecMesh employs a lightweight, multi-head speculative decoding scheme to predict multiple tokens in parallel within a single forward pass, thereby accelerating inference. We further propose a verification and resampling strategy: the backbone model verifies each predicted token and resamples any tokens that do not meet the quality criteria. In addition, we propose a distillation strategy that trains the lightweight decoding heads by distilling from the backbone model, encouraging their prediction distributions to align and improving the success rate of speculative predictions. Extensive experiments demonstrate that our method achieves a $1.7\times$ speedup without sacrificing generation quality. Our code will be released.

## 1 Introduction

Triangular meshes constitute the foundation of 3D representation and are extensively employed across industries, including virtual reality, gaming, animation, and product design. High-quality meshes exhibiting precise topology are essential for downstream tasks, such as mesh editing, skeletal rigging, texture mapping, and animation. However, constructing meshes with fine-grained topology remains a labor-intensive endeavor that requires substantial design effort, thus impeding the advancement of 3D content creation. Recent works employ auto-regressive architectures Siddiqui et al. (2024); Chen et al. (2024a;b); Weng et al. (2025); Zhao et al. (2025a); Liu et al. (2025) for

token-based mesh generation, they directly generate mesh vertices and faces while demonstrating the capacity to produce topologically precise meshes. However, the auto-regressive paradigm incurs high inference latency: existing auto-regressive mesh generation models depend on next-token predictions, requiring thousands to tens of thousands of forward passes to produce a single 3D mesh.

We draw inspiration from Speculative Decoding Leviathan et al. (2023); Chen et al. (2023) in efficient LLM inference, which typically employs a draft model with significantly fewer parameters than the original. The draft model generates candidate tokens, which the original model then verifies—enabling near-draft-model generation speed while preserving the original model's generation quality. However, draft models must satisfy stringent criteria: their parameter count must be sufficiently constrained to facilitate accelerated inference, and their predictions must closely align with the distribution of the original model. Consequently, deriving such draft models remains a challengeChen et al. (2023); Leviathan et al. (2023). On the other hand, we note that, unlike auto-regressive language models which frequently employ larger vocabularies to enhance expressiveness Tao et al. (2024); Huang et al. (2025), existing auto-regressive mesh generation models typically utilize efficient, compressed representations to minimize vocabulary size. Table 1 summarizes these disparities. This discrepancy motivates us to explore a more lightweight decoding design to obtain the probability distribution over the vocabulary.

To this end, we introduce XSpecMesh, a novel framework that accelerates auto-regressive mesh generation models while preserving generation quality. The framework implements multi-head speculative decoding to accelerate inference: multiple lightweight decoding heads simultaneously predict a sequence of subsequent tokens in a single forward pass. These decoding heads leverage cross-attention mechanisms with the generation conditions to enhance prediction accuracy. Furthermore, we introduce a verification and resampling strategy to evaluate candidate tokens predicted by the decoding heads, resampling those that fail to meet quality criteria, thereby ensuring that output quality remains uncompromised. Finally, we employ backbone distillation training to encourage the decoding heads' predictive distributions to approximate that of the backbone model, allowing the backbone to accept their predictions. Figure **??** illustrates the differences between our framework and previous works.

To the best of our knowledge, XSpecMesh is the first method that accelerates inference in auto-regressive mesh generation models without sacrificing generation quality. Our contributions can be summarized as follows:

- We propose XSpecMesh, a method to accelerate auto-regressive mesh generation models without compromising generation quality, by employing multiple cross-attention speculative decoding heads for multi-token prediction.
- We develop a verification and resampling strategy that, within a single forward pass, employs the backbone model to verify candidate tokens and resample those that do not meet predefined quality criteria, thereby ensuring uncompromised generation quality.
- We further introduce a distillation strategy to train decoding heads, aligning their prediction distribution with the backbone model to improve the success rate of speculative predictions.
- Extensive experiments demonstrate that our method accelerates inference without sacrificing generation quality, achieving a $1.7\times$ speedup.

## 2 RELATED WORKS

### 2.1 3D MESH GENERATION

Due to the complexity of direct mesh generation, many 3D synthesis methods utilize intermediate representations—such as voxels Wu et al. (2016); Wang et al. (2017), point clouds Luo & Hu (2021); Jun & Nichol (2023); Qi et al. (2017), or implicit fields Chen & Zhang (2019); Park et al. (2019)—to avoid modeling meshes directly. Representative approaches include optimizing 3D representations within pretrained 2D diffusion models via score-distillation sampling (SDS) Poole et al. (2022); Wang et al. (2023); Zhu et al. (2023); Li et al. (2023); Lin et al. (2023); Tang et al. (2023); Yi et al. (2024), 3D transformer models Hong et al. (2023); Tang et al. (2024a); Xu et al. (2024), and the recent 3D latent diffusion models Zhang et al. (2024); Xiang et al. (2025); Hunyuan3D et al. (2025); Wu et al. (2024); Zhao et al. (2025b) that achieve high-quality shape generation. These

| Method | BPT | DeepMesh | LLaMa 3 | Qwen3 |
|--------|-----|----------|---------|-------|
| Vocab Size | 5120 | 4736 | 128K | 152K |

Table 1: **The difference in vocabulary size between auto-regressive mesh generation models and language models.** Language models Grattafiori et al. (2024); Yang et al. (2025) tend to use larger vocabularies to enhance expressiveness, whereas auto-regressive mesh generation models favor efficient compressed representations to reduce vocabulary size.

approaches typically apply Marching Cubes Lorensen & Cline (1998) in post-processing to extract meshes, frequently introducing topological artifacts. In contrast, MeshGPT Siddiqui et al. (2024), which integrates VQ-VAE Van Den Oord et al. (2017) with a transformer Vaswani et al. (2017) for auto-regressive mesh generation, produces high-quality topological meshes; however, it is confined to low-polygon meshes and single-category shapes. A subsequent series of auto-regressive mesh generation methods Chen et al. (2024b;c); Tang et al. (2024b); Hao et al. (2024); Chen et al. (2024a); Liu et al. (2025), has demonstrated the ability to synthesize topologically precise meshes, BPT Weng et al. (2025) and DeepMesh Zhao et al. (2025a) further scale auto-regressive mesh generation to large datasets through efficient tokenization schemes. However, the intrinsic latency of the auto-regressive paradigm hinders its applicability. In this paper, we therefore propose a novel method to accelerate auto-regressive mesh generation while preserving generation quality.

## 2.2 ACCELERATION OF AUTO-REGRESSIVE MODEL

Various strategies have been proposed to accelerate auto-regressive language models: weight pruning methods Frantar & Alistarh (2023); Sanh et al. (2020) eliminate redundant parameters to decrease computational load; quantization techniques Frantar et al. (2022); Xiao et al. (2023) convert models into low-bit representations to cut memory and compute overhead; and sparsity-based approaches Fedus et al. (2022); Fu et al. (2024a) reduce activation computations to improve efficiency. Nonetheless, these methods retain the conventional auto-regressive, token-by-token decoding paradigm. An alternative research direction Gloeckle et al. (2024); Fan et al. (2025); Cai et al. (2024); Wang et al. (2025) attempts to predict multiple tokens in a single forward pass to reduce iterative decoding steps. The Speculative Decoding approaches Chen et al. (2023); Sun et al. (2023); Leviathan et al. (2023) employ a draft model to generate tokens rapidly, then verify them with the original model to preserve generation quality. Certain efforts target acceleration of auto-regressive image synthesis: SJD Teng et al. (2024) integrates Speculative Decoding with Jacobi decoding, whereas ZipAR He et al. (2024) exploits local sparsity for parallel token generation. To date, these acceleration studies have focused predominantly on language and image generation domains, with auto-regressive mesh generation remaining insufficiently explored.

## 3 PRELIMINARY

### 3.1 AUTO-REGRESSIVE MESH GENERATION

An auto-regressive mesh generation framework comprises three fundamental components: a discrete mesh serialization method Chen et al. (2024b); Weng et al. (2025) that converts vertices and faces into a token sequence; a transformer-based auto-regressive generator that, conditioned on input prompts, sequentially predicts each subsequent token to generate the token sequence; a deserialization method that reconstructs the 3D mesh vertices and faces from the generated sequence.

Auto-regressive models employ causal masking during training, so that, for a given sequence $x_{0:n}$, the model can perform, in a single forward pass, simultaneous computations of the predictive distributions for positions $1, 2, \ldots, n+1$:

$$p_1(x|x_0), \ p_2(x|x_{0:1}), \ \ldots, \ p_{n+1}(x|x_{0:n}). \tag{1}$$

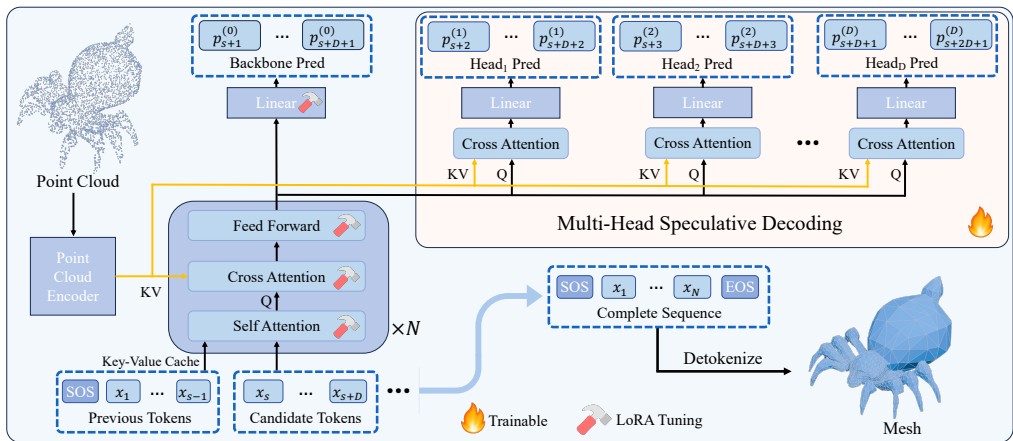

Figure 2: **Overview of our method.** Left: A pretrained transformer-based auto-regressive mesh generation model, fine-tuned with LoRA. Top-right: The transformer's final hidden layer is decoded by $D$ cross-attention decoding heads, the $d$-th head predicts the $(d+1)$-th next token. Bottom-right: The complete generated token sequence is detokenized to produce the mesh.

For each position $i$, with corresponding target label $y_i$, the model is trained by minimizing the cross-entropy loss:

$$\mathcal{L} = \sum_i -\log p_i(y_i). \tag{2}$$

This property also means that, at inference time, by evaluating $p_{i+1}(x|x_{0:i})$, one can determine whether a candidate token $x_{i+1}$ aligns with the model's learned distribution. Our method leverages this property to accelerate generation without compromising quality.

## 4 METHOD

Our method aims to accelerate auto-regressive mesh generation models without compromising generation quality. We propose multi-head speculative decoding, in which multiple lightweight cross-attention decoding heads concurrently predict subsequent tokens, thereby accelerating the sequence generation process (Sec 4.1). Since these decoding heads' predictions may be imprecise, we employ the backbone model's robust prior to verify outputs—rejecting and resampling at the first invalid token—to guarantee generation quality (Sec 4.2). To enhance acceptance of decoding heads' proposals, we distill backbone knowledge into these heads during training, aligning their output distributions with the backbone's (Sec 4.3). Figure 2 provides an overview of our method.

### 4.1 MULTI-HEAD SPECULATIVE DECODING

Auto-regressive models exhibit excellent generation quality, however, their inference relies on sequential, token-by-token generation, leading to high latency. To alleviate this bottleneck, we introduce multi-head speculative decoding. In auto-regressive mesh generation models, the vocabulary size is considerably smaller than that of LLMs (Table 1), resulting in a relatively simple decoding process. Therefore, we propose a more efficient approach that employs multiple lightweight decoding heads to process the transformer's final hidden layer and predict subsequent tokens.

Specifically, the backbone model comprises $N$ transformer blocks, each containing: a self-attention layer, a cross-attention layer for injecting the generation condition $c$, and a feed-forward network. Let $s$ denote the current sequence position, and assume tokens $x_0$ through $x_{s-1}$ are stored in the key–value cache. Denote the layer-0 hidden state as $h_s^0 = x_s$. Then, for $l = 0, 1, \ldots, N-1$, the $(l+1)$-th hidden state is computed as $h_s^{l+1} = \text{block}^l(h_s^l, c)$. Define the final hidden state as $h_s = h_s^N$. The backbone model subsequently decodes $h_s$ through a linear layer $W^{(0)}$ to yield the

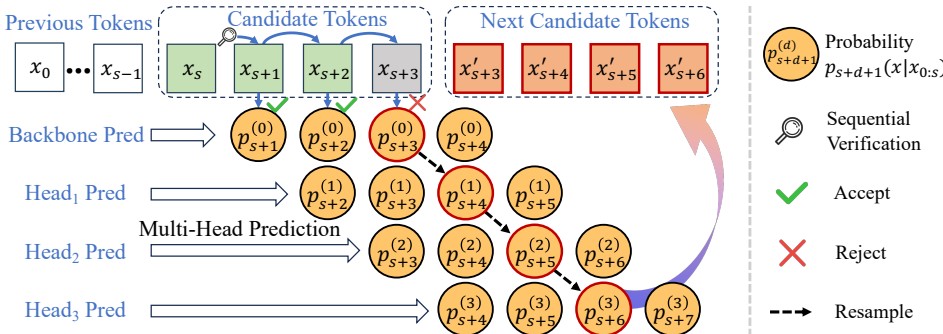

Figure 3: **Verification and resampling.** The figure uses $D = 3$ as an example to illustrate the process. Each candidate token sampled in a forward pass must be verified by the backbone model: if $p_i^{(0)}(x_i) > \delta$, token $x_i$ is accepted and verification proceeds to the next token, until the first token $x_{i'}$ that fails the verification condition. At that point, new tokens are resampled from the distributions $p_{i'}^{(0)}, p_{i'+1}^{(0)}, \ldots, p_{i'+D}^{(0)}$ to form the candidate tokens for the next iteration.

probability distribution for the next token at position $s + 1$:

$$p_{s+1}^{(0)} = \text{softmax}(W^{(0)} \cdot h_s). \tag{3}$$

Given the generation condition $c$, we employ multiple cross-attention decoding heads to decode $h_s$, with the $d$-th decoding head predicting the token at position $s + d + 1$:

$$p_{s+d+1}^{(d)} = \text{softmax}(W^{(d)} \cdot \text{CrossAttn}^{(d)}(h_s, c)). \tag{4}$$

Compared to decoding via an MLP, using a cross-attention mechanism allows the decoding heads to better align with the input conditional features, thereby improving the accuracy of subsequent-token predictions. Finally, we sample from probability distributions $p_{s+1}^{(0)}, p_{s+2}^{(1)}, \ldots, p_{s+D+1}^{(D)}$ to generate the tokens $x_{s+1}, x_{s+2}, \ldots, x_{s+D+1}$.

## 4.2 VERIFICATION AND RESAMPLING

After generating the next $D + 1$ tokens from position $s$ via the backbone model and $D$ decoding heads, the straightforward approach is to append these tokens to the existing sequence and resume prediction at position $s + D + 1$. However, due to potential inaccuracies of the decoding heads, this strategy can drastically degrade the generated sequence's quality. We therefore propose a verification strategy that leverages the backbone model to simultaneously verify and resample tokens in a single forward pass.

Specifically, we leverage the backbone model's prior judgment to determine whether to accept tokens predicted by the decoding heads. Let $s$ denote the current accepted sequence position. In a single forward pass, the backbone model employs causal masking on the sequence $x_{s:s+D}$ to obtain $p_{s+1:s+D+1}^{(0)}$, and based on this probability distribution, partially accepts a prefix $x_{s:s^*-1}$. Subsequently, with the backbone model and $D$ decoding heads, we resample tokens at positions $s^*$ to $s^* + D$. We apply a probability-threshold-based criterion: a token $x_i$ is accepted if $p_i^{(0)}(x_i) > \delta$. Figure 3 provides a detailed illustration of this process.

By verifying with the backbone model and sampling with multiple decoding heads, we reduce the number of forward passes through the backbone model while preserving generation quality, thus speeding up the overall generation process. Algorithm 1 presents a detailed description of the multi-head speculative decoding procedure.

## 4.3 BACKBONE DISTILLATION TRAINING

Analogous to Speculative Decoding, in which the draft model's output distribution must closely match that of the original model, our framework requires the decoding heads' output distributions to

---

**Algorithm 1** Multi-Head Speculative Decoding

---

**Input**: Condition $c$, Backbone Model $\mathcal{M}$, Multi-Head Speculative Decoder $\{\mathcal{H}_i\}_{i=1}^{D}$
**Output**: Mesh Sequence $x_{0:i_{\text{EOS}}}$

1: Let $x_0 \leftarrow \text{SOS}$, $x_{1:D} \sim U(0, V)$, $s \leftarrow 0$.
2: **while** $s < L_{max}$ and $x_{0:s} \neq \text{EOS}$ **do**
3:     $p_{s+1:s+D+1}^{(0)}$, $h_{s:s+D} \leftarrow \mathcal{M}(x_{s:s+D}, c)$                 ▷ forward with causal mask
4:     **for** $i = 1$ to $D$ **do**
5:         $p_{s+1+i:s+D+1+i}^{(i)} \leftarrow \mathcal{H}_i(h_{s:s+D}, c)$
6:     **end for**
7:     $s^* \leftarrow s + 1$
8:     **while** $s^* < s + D + 1$ and $p_{s^*}^{(0)}(x_{s^*}) > \delta$ **do**
9:         $s^* \leftarrow s^* + 1$                                 ▷ verify and accept
10:     **end while**
11:     $x_{s^*:s^*+D} \leftarrow \text{sample}(p_{s^*}^{(0)}, p_{s^*+1}^{(1)}, \ldots, p_{s^*+D}^{(D)})$   ▷ resample from the first rejected position $s^*$
12:     $s \leftarrow s^*$
13: **end while**
14: **return** $x_{0:i_{\text{EOS}}}$

---

align with the backbone model's distribution to ensure acceptance of their predictions. To this end, we distill the backbone model to train decoding heads. We sample point clouds from the dataset and employ the backbone model to generate sequences, which serves as the ground truth labels $y_{0:n}$ for decoding heads training, We train the $d$-th decoding head using the cross-entropy loss:

$$\mathcal{L}_d = \sum_s -\log p_{s+d+1}^{(d)}(y_{s+d+1}). \tag{5}$$

With increasing $d$, the accuracy of the $d$-th decoding head declines, potentially causing gradient instability. To mitigate this issue, we introduce a weighting function $w(d)$, which decreases as $d$ increases. Accordingly, the overall loss for the $D$ decoding heads is formulated as follows:

$$\mathcal{L}_{\text{mhd}} = \sum_{d=1}^{D} w(d) \cdot \mathcal{L}_d. \tag{6}$$

Following decoding heads training, they are deployed for inference acceleration. Empirical evaluation, however, indicates that the speed-up benefits are modest. This limitation arises because the backbone model is optimized under a next-token prediction paradigm, making direct decoding of subsequent tokens from the hidden state $h_s$ infeasible. To mitigate this issue, we fine-tune the backbone model's linear layer via LoRA Hu et al. (2022), enabling the decoding heads to more effectively derive multiple subsequent token predictions from $h_s$. Training proceeds in two stages. In the first stage, we train only the decoding heads while freezing the backbone model to prevent unstable gradients from the decoding heads in the early training stage from affecting the backbone model. In the second stage, we jointly train both the decoding heads and LoRA. Furthermore, we integrate the backbone model's prediction loss $\mathcal{L}_{\text{backbone}} = \sum_s -\log p_{s+1}^{(0)}(y_{s+1})$ into the overall objective with a substantial weighting factor $\lambda$, ensuring gradients from the decoding heads do not diverge the backbone distribution from its original form. The loss function for the second stage is formulated as follows:

$$\mathcal{L}_{\text{total}} = \lambda \mathcal{L}_{\text{backbone}} + \mathcal{L}_{\text{mhd}}. \tag{7}$$

Although during training LoRA introduces two low-rank matrices $A$ and $B$ to each original linear layer weight matrix $W$, at inference time these LoRA weights can be merged with the original weights $W_{\text{origin}}$ via a simple preprocessing step to form the merged weight $W_{\text{merge}} = W_{\text{origin}} + AB$. Therefore, introducing LoRA incurs no additional computational overhead. Upon fine-tuning the backbone model with LoRA, the decoding heads are able to accurately predict subsequent tokens, significantly increasing the decoding speed.

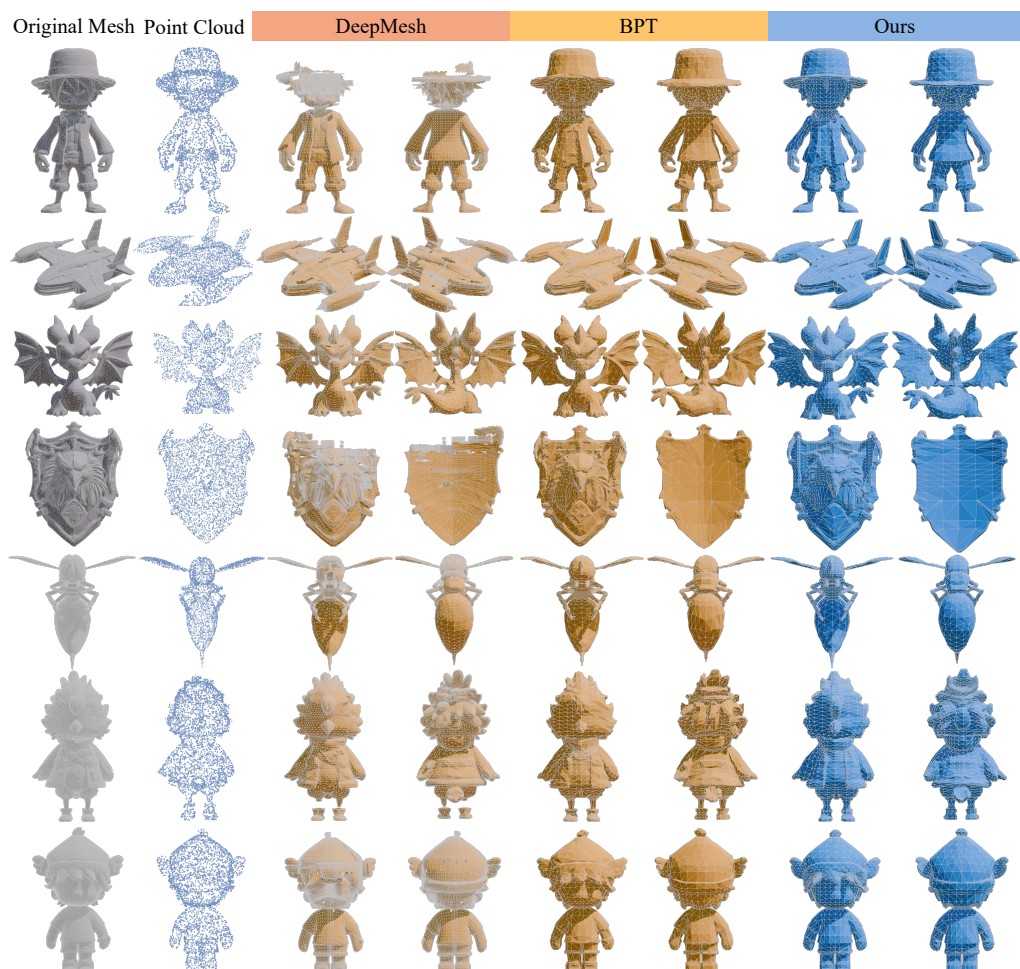

Figure 4: **Comparison of our method with BPT and DeepMesh.**

## 5 EXPERIMENTS

### 5.1 EXPERIMENT SETTINGS

**Implementation Details.** We adopt BPT Weng et al. (2025) as our base model: an auto-regressive mesh generation model pretrained on a large-scale, high-quality dataset. We train on a subset of Objaverse Deitke et al. (2023) containing approximately 10K shapes. In the first stage, we train only the decoding heads, setting the loss weight for the $d$-th decoding head to $w(d) = 0.8^d$. In the second stage, we jointly train the LoRA adapters and the decoding heads; to prevent the backbone model's distribution from drifting, we assign a relatively large weight $\lambda = 50$ to the backbone loss. See the Appendix for more details.

**Baselines.** We compare our method against the base model BPT and another state-of-the-art auto-regressive mesh generation model, DeepMesh Zhao et al. (2025a). Since DeepMesh has only released a 0.5B-parameter configuration, we use this version for evaluation.

**Metrics.** We follow the evaluation procedure of previous work Weng et al. (2025); Zhao et al. (2025a); Liu et al. (2025), and generate 200 test meshes via the generation model Xiang et al. (2025); Zhao et al. (2025b) (see the Appendix for more details). We uniformly sample 1,024 points from the surfaces of ground-truth and generated meshes, computing Chamfer Distance (CD) and Hausdorff Distance (HD) as objective quality metrics. Additionally, a user study (US) is conducted to capture subjective assessments. For speedup evaluation, we follow the methodology of previous work Fu et al. (2024b); Xia et al. (2022); Chen et al. (2023) and define the Step Compression Ratio

| Method | CD ↓ | HD ↓ | US ↑ | Avg. Lat. ↓ |
|---|---|---|---|---|
| DeepMesh | 0.1323 | 0.2648 | 27% | 979.6s |
| BPT | 0.1165 | 0.2223 | 37% | 257.6s |
| Ours | 0.1168 | 0.2261 | 36% | **151.4s** |

Table 2: **Quantitative comparison with other methods.** Avg. Lat. denotes the average latency to generate the complete mesh sequence (measured on the RTX 3090).

| Configuration | CD ↓ | HD ↓ | SCR ↑ | Step Latency ↓ | Speedup ↑ |
|---|---|---|---|---|---|
| **A** BPT | 0.1165 | 0.2223 | 1.000 | 40.51ms | 1.00× |
| **B** *w.* MLP Decoder | 0.1195 | 0.2241 | 1.181 | 44.89ms | 1.07× |
| **C** *w.* MLP Decoder & LoRA | 0.1267 | 0.2485 | 1.909 | 44.92ms | 1.65× |
| **D** *w.* CA Decoder | 0.1167 | 0.2229 | 1.334 | 47.81ms | 1.13× |
| **E** *w.* CA Decoder & LoRA (Ours) | 0.1168 | 0.2261 | 2.021 | 47.83ms | **1.71×** |

Table 3: **Ablation across different configurations.** The Cross-Attention decoding heads incorporate generation conditions, achieving excellent performance in both generation quality and speedup.

as: $\text{SCR} = \frac{\text{\#generated and accepted tokens}}{\text{\#decoding steps}}$, where a decoding step denotes the process of verifying and decoding multiple tokens in a single forward pass. Since we introduced additional decoding heads, we measured the latency of a single decoding step (Step Latency) on an RTX 3090. Finally, we computed the actual speedup ratio (Speedup) based on SCR and Step Latency.

## 5.2 QUALITATIVE RESULTS

We perform a qualitative comparison of our method against established baselines, presenting several challenging examples in Figure 4. Although DeepMesh can generate higher-resolution meshes, its truncated-window training induces context loss, resulting in fragmented meshes. In contrast, BPT yields more consistent generation results, while our approach achieves shape and topological fidelity comparable to BPT.

## 5.3 QUANTITATIVE RESULTS

Table 2 summarizes the results of our quantitative comparison. DeepMesh is capable of generating high-resolution meshes, which has earned it a certain level of popularity in user study. However, owing to its propensity to produce fragmented and incomplete meshes, DeepMesh exhibits higher CD and HD values. By contrast, the results generated by BPT demonstrate greater consistency. Since our method produces results highly similar to BPT, the corresponding CD and HD metrics are comparable. Moreover, in the user study where methods were anonymized, participants were unable to differentiate between our method's outputs and those of BPT, yielding comparable survey scores (see the Appendix for more details). Overall, our method matches the baseline BPT in generation quality while significantly reducing complete mesh sequence generation latency.

## 5.4 ABLATION STUDY

**Decoding head architectures and training strategies.** We first compared the quality of the generated shapes and the achieved speed-up under different decoding head architectures and training strategies: A. Baseline model: BPT; B. MLP decoding heads, training only the first-stage decoding heads; C. MLP decoding heads, first training the first-stage decoding heads, then jointly training LoRA adapters and decoding heads in a second stage; D. Cross-attention decoding heads, training only the first-stage decoding heads; E. Cross-attention decoding heads, first training the first-stage decoding heads, then jointly training LoRA adapters and decoding heads in a second stage.

Table 3 presents the evaluation results for different configurations. Compared to the MLP decoding heads, the cross-attention decoding heads, despite incurring higher step latency, more effectively integrate conditional information into the generation process, thereby yielding more accurate pre-

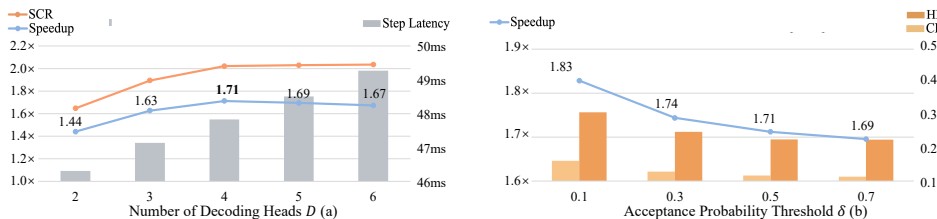

Figure 5: Ablation of the number of decoding heads $D$ and the acceptance probability threshold $\delta$.

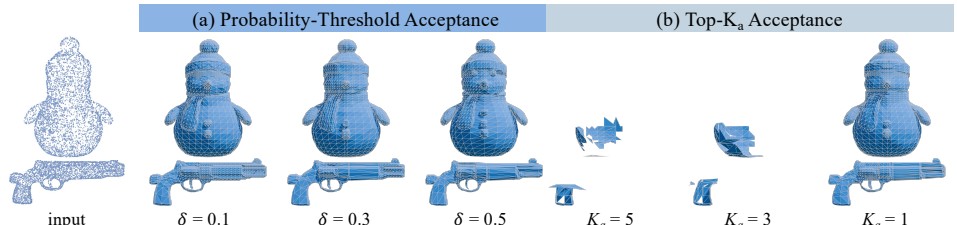

Figure 6: **Comparison of Probability-Threshold and Top-$K_a$ Acceptance.**

dictions of subsequent tokens and consequently improving the SCR. After two-stage joint training with LoRA, the MLP decoding heads also achieve a comparably high speedup; however, their generation quality deteriorates to some extent. This degradation stems mainly from (1) Joint training with LoRA aligns the prediction distributions of the decoding heads with those of the backbone model, thereby increasing the backbone's propensity to accept the decoding head's outputs, and (2) the MLP decoding heads' predictions, lacking injected conditional information, produce some inaccuracies that the backbone model still accepts, thereby compromising overall quality. In contrast, the configuration integrating cross-attention decoding heads with LoRA joint training better aligns multi-token predictions with the generation conditions, achieving the highest speedup while also outperforming MLP decoding heads on both CD and HD metrics.

**Number of decoding heads.** Increasing the number of decoding heads raises SCR but also increases step latency. As shown in Figure 5(a), we present SCR and step latency for various numbers of decoding heads and subsequently compute speedup. At $D = 4$, speedup peaks at $1.71\times$.

**Verification criterion.** We use a threshold $\delta$ as the acceptance condition: a token $x_i$ is accepted if $p_i^{(0)}(x_i) > \delta$. As the hyperparameter $\delta$ increases, the criterion becomes stricter, leading to lower speedup but improved generation quality. Figure 5(b) illustrates the impact of varying $\delta$ on speedup, CD, and HD. At $\delta = 0.5$, our method achieves an optimal trade-off between speedup and generation quality, delivering substantial acceleration while preserving quality comparable to the baseline model. Furthermore, we compare two acceptance criteria—Probability-Threshold Acceptance and Top-$K_a$ Acceptance (a token $x_i$ is accepted if $x_i$ is among the top-$K_a$ tokens of $p_i^{(0)}$)—and present the results in Figure 6. For top-$K_a$ acceptance, a long-tail effect arises: certain candidate tokens within the top-$K_a$ may exhibit exceedingly low probabilities yet be accepted, severely degrading generation quality. Only for $K_a = 1$ does the model generate a reasonable shape. By contrast, probability-threshold acceptance demonstrates greater stability, yielding satisfactory results for thresholds between 0.1 and 0.5.

**Sampling strategies.** We compare two sampling strategies: Independent Sampling and Top-$K_s$ Probability-Tree Sampling, see the Appendix for more details.

## 6 CONCLUSION

We propose XSpecMesh, which accelerates auto-regressive mesh generation models by using multiple cross-attention decoding heads for multi-token prediction. By employing multi-head speculative decoding with a verification and resampling strategy, our method achieves a $1.7\times$ speedup over the base model while preserving generation quality.

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

# A  APPENDIX

## A.1  THE USE OF LARGE LANGUAGE MODELS (LLMS)

All technicalcontributions, including the methodology, equations, and results, are solely the work of the authors.

## A.2  IMPLEMENTATION DETAILS

Training was performed on two NVIDIA A800 GPUs and took approximately eight hours. We used AdamW as the optimizer ($\beta_1 = 0.9, \beta_2 = 0.99$). To improve training stability, we applied global norm clipping to the gradients, limiting their overall norm to within 1.0. The training procedure comprised two stages. During stage one, we trained only the decoding head, employing a cosine learning rate schedule decaying from $5 \times 10^{-4}$ to $5 \times 10^{-5}$ over 30 epochs. Subsequently, we applied LoRA to fine-tune the backbone, jointly training both modules for 10 epochs with a cosine learning rate schedule decaying from $1 \times 10^{-4}$ to $1 \times 10^{-5}$. We set the LoRA rank to 16 and alpha to 32.

## A.3  TEST DATASET

Our test data was generated from the generation model Xiang et al. (2025); Zhao et al. (2025b) and covers a rich and diverse set of shapes. Moreover, we categorized the shapes in the test dataset into three different difficulty levels: level-0, level-1, and level-2. (1) level-0: Simple shapes with minimal detail. (2) level-1: Relatively complex shapes with a certain amount of detail. (3) level-2: Challenging shapes featuring a rich array of details. In the entire test dataset, level-0 accounts for approximately 20%, level-1 for 40%, and level-2 for another 40%. We showcase a subset of the shapes from the test set in Figure 7.

## A.4  ABLATION OF SAMPLING STRATEGIES

We conducted a study of sampling strategies by comparing two methods: Independent Sampling (IS) and Top-$K_s$ Probability Tree Sampling (PTS).

**Independent Sampling.** Samples are drawn independently from each decoding head's probability distribution $p^{(d)}$, with token probabilities serving as sampling weights.

**Top-$K_s$ Probability Tree Sampling.** For each layer's distribution $p^{(d)}$, the Top-$K_s$ tokens by probability are selected to recursively construct a probability tree. Denote the probabilities of the Top-$K_s$ tokens at layer $d$ by $\{m_{i_{d,k}}^{(d)}\}_{k=1}^{K_s}$. The weight of a path from the root to a leaf is then computed as $\prod_{d=1}^{D} m_{i_{d,k}}^{(d)}$. To constrain tree-construction complexity, branches with cumulative weights below $1 \times 10^{-5}$ are pruned. Complete paths are then sampled according to their accumulated path probabilities.

Compared to IS, Top-$K_s$ PTS improves the step compression ratio (SCR) by considering combinations among sampled tokens, but because each iteration requires building a search tree—incurring additional overhead—it does not achieve a higher speedup. The results are shown in Table 4.

## A.5  USER STUDY

We randomly selected 70 participants to complete a questionnaire as a subjective metric. Each questionnaire comprised 20 cases, resulting in 1,400 responses in total. Outputs from DeepMesh, BPT, and our method were randomly shuffled and anonymized to ensure fairness. For each case, participants were instructed to holistically evaluate both the generated shape and wireframe topology, then select the most favorable result. Owing to its tendency to generate fragmented and incomplete meshes, DeepMesh received relatively fewer votes. By contrast, participants struggled to distinguish between BPT and our method, resulting in nearly identical vote counts for these two approaches.

| Method | SCR ↑ | Step Latency ↓ | Speedup ↑ |
|---|---|---|---|
| IS | 2.021 | 47.83ms | 1.71× |
| PTS($K_s = 2$) | 2.030 | 48.06ms | 1.71× |
| PTS($K_s = 3$) | 2.033 | 48.47ms | 1.69× |
| PTS($K_s = 4$) | 2.036 | 49.08ms | 1.68× |

Table 4: **Comparison of Independent Sampling (IS) and Top-$K_s$ Probability Tree Sampling (PTS).** Top-$K_s$ PTS achieves a higher SCR, but due to the overhead of building the search tree at each iteration, its actual speedup is slightly lower than that of Independent Sampling.

### A.6 ANALYSIS OF QUALITATIVE AND QUANTITATIVE COMPARISONS

While DeepMesh can produce meshes with greater face counts and finer details, it requires substantially longer token sequences. To mitigate this, DeepMesh was trained with a truncated attention window and a maximum inference context size of 9,000 tokens—design decisions that result in fragmented meshes, as illustrated by the red boxes in Figure 4 of the main text. Furthermore, DeepMesh frequently produces meshes that are overly dense yet incomplete, as evidenced in rows 1 and 4. These shortcomings inflate its CD and HD metrics and diminish its user-study vote share.

### A.7 LoRA INSTEAD OF FULL PARAMETERS TUNING

We fine-tune the backbone model using LoRA rather than full-parameter fine-tuning. Compared to full-parameter tuning, LoRA is more training-efficient and converges faster. Equally important, LoRA effectively prevents distribution drift in the backbone model. Since our method relies on the backbone to verify multiple candidate tokens, its predictions are critical to generation quality. With full-parameter fine-tuning, gradients originating from the decoding heads can cause certain backbone parameters to drift significantly, harming sampling quality. By contrast, LoRA applies low-rank update matrices to the model; these low-rank updates curb any severe parameter drift induced by decoding-head gradients during training, thus preserving generation quality.

### A.8 MORE RESULTS

We further collected more examples, and displayed the generated results of BPT and our method in Figures 8 and 9. In these challenging cases, our approach is capable of producing meshes with shape

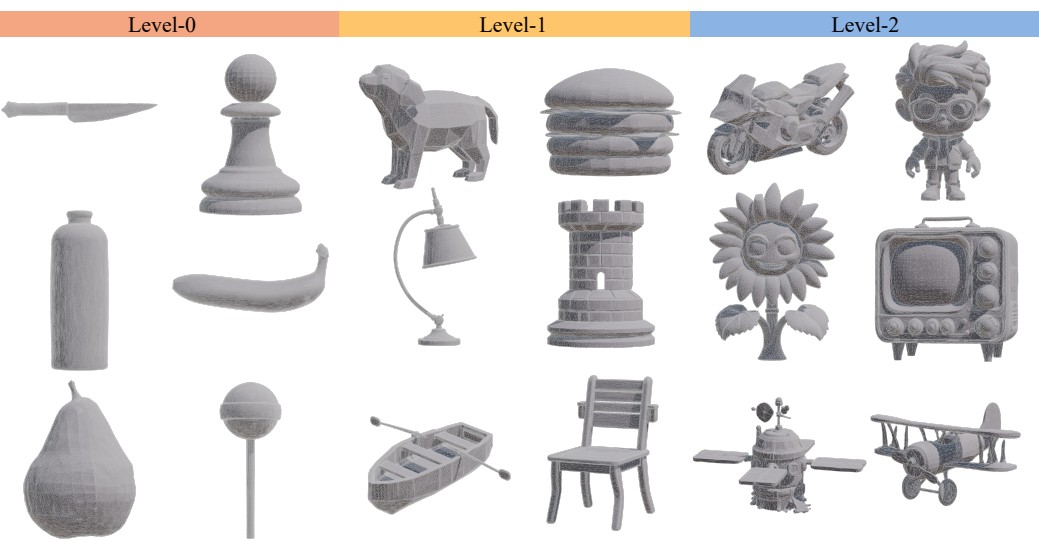

Figure 7: **A subset of examples from the test dataset.** Our test dataset contains a rich variety of shapes and is divided into three different difficulty levels: level-0, level-1, and level-2.

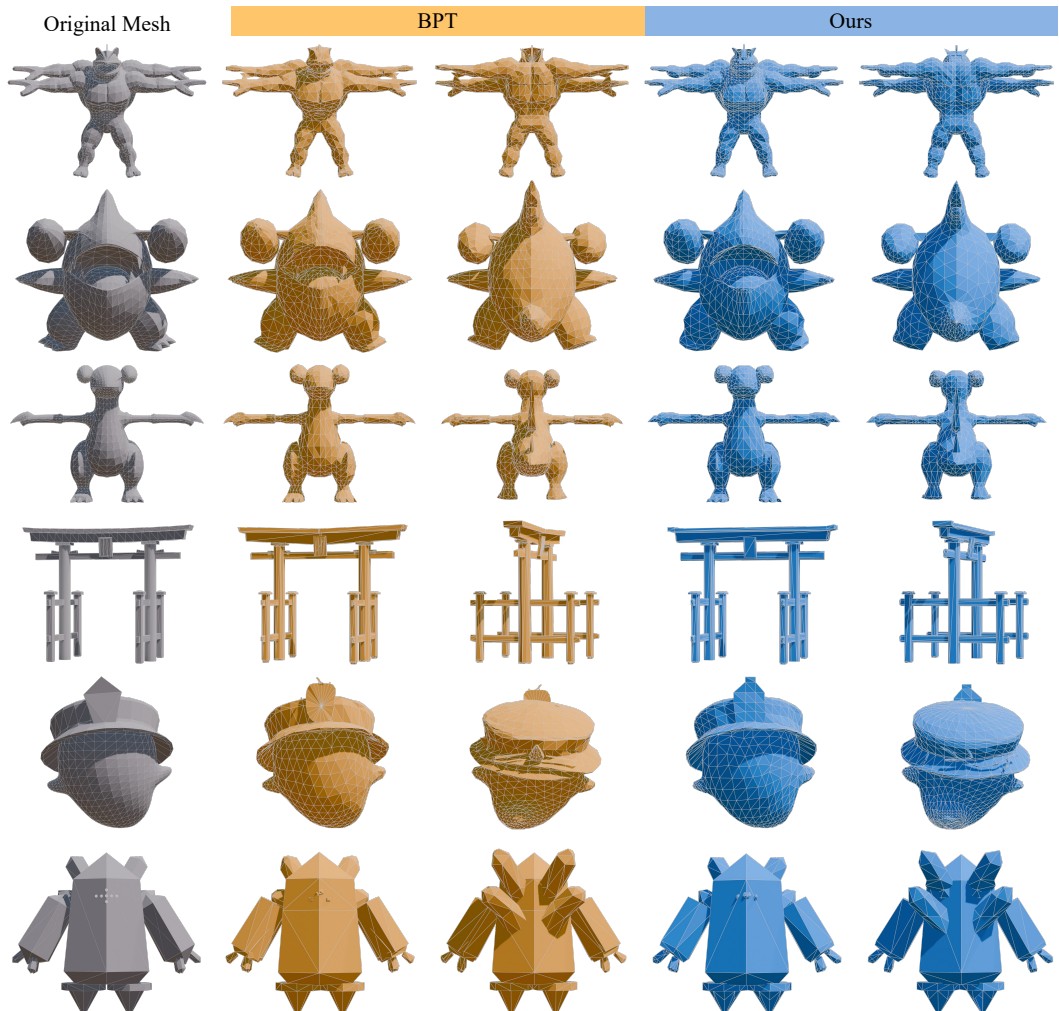

Figure 8: **Additional generation results of our method versus BPT.** Our acceleration method, built upon BPT, substantially accelerates generation while preserving BPT's shape and topological fidelity.

and topology quality comparable to that of the base model BPT, while significantly accelerating the generation speed.

## A.9 VISUALIZATION OF FAILURE CASES

Since our method builds on the base model, it exhibits the same failure modes as BPT on out-of-distribution data. Figure 10 presents visualizations of several such failure cases.

## A.10 LIMITATION

Although our method significantly accelerates the base model's generation speed without sacrificing output quality, we still employ the base model as the backbone and use it to validate candidate tokens to ensure generation quality; consequently, the performance of our approach remains constrained by that of the base model.

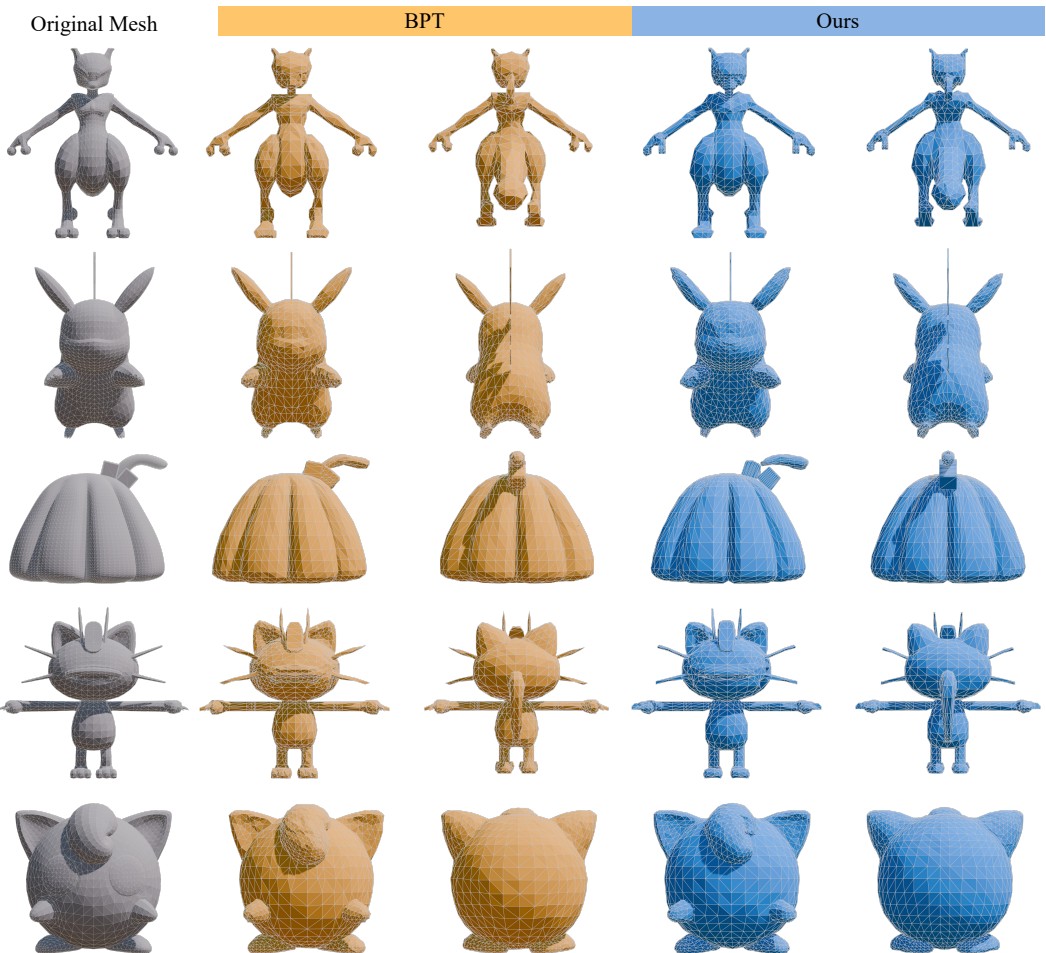

Figure 9: **Additional generation results of our method versus BPT.** Our acceleration method, built upon BPT, substantially accelerates generation while preserving BPT's shape and topological fidelity.

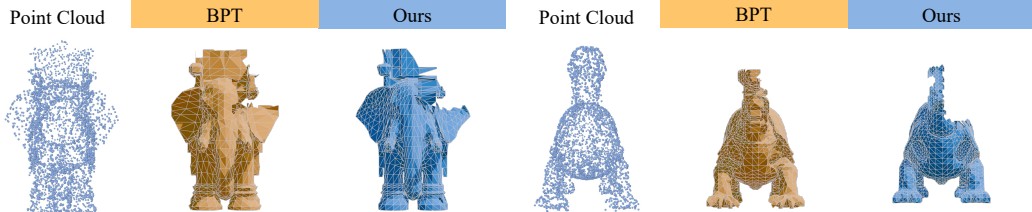

Figure 10: **Visualization of failure cases.**

## A.11 GENERALITY

To further demonstrate the generality of our method, we also apply it to DeepMesh for acceleration. Using the same truncated-window training strategy as DeepMesh, the results are shown in Table 5.

Since DeepMesh adopts a hourglass transformer, its hierarchical structure introduces technical challenges for maintaining the KV cache. Specifically, the hourglass transformer performs hierarchical downsampling and upsampling over the sequence. We maintain a separate KV cache for each layer and record the valid KV cache positions (i.e., positions of tokens that have already been verified). Each time the KV cache is used, only the valid positions are accessed. When a token is verified,

| Method | CD ↑ | HD ↓ | SCR ↑ | Speedup ↑ |
|---|---|---|---|---|
| DeepMesh | 0.1323 | 0.2648 | 1.000 | 1.00× |
| DeepMesh+ours | 0.1378 | 0.2767 | 2.184 | 1.52× |
| BPT | 0.1165 | 0.2223 | 1.000 | 1.00× |
| BPT+ours | 0.1168 | 0.2261 | 2.021 | 1.71× |

Table 5: **The acceleration results of applying our method to different foundation models.**

the valid positions in the KV cache are updated layer by layer. In addition, we employ hierarchical absolute position writes to the KV cache, meaning that at each forward pass, the newest values are written into the corresponding KV cache positions, overwriting previously invalid entries.

