# OpenReview forum: "Quality-Preserving Auto-Regressive Mesh Generation Acceleration via Multi-Head Speculative Decoding"
_ICLR.cc/2026/Conference — ICLR 2026 Conference Withdrawn Submission_

### Official Review · Reviewer_HMRD · 2025-10-18

**Soundness:** 3
**Presentation:** 3
**Contribution:** 2
**Rating:** 4
**Confidence:** 5

**Summary:**

This paper introduces XSpecMesh, a method to accelerate the inference speed of auto-regressive mesh generation models. The core problem addressed is the high latency caused by the sequential, token-by-token generation process. The proposed solution adapts the concept of speculative decoding from large language models. Instead of a separate draft model, XSpecMesh attaches multiple lightweight, cross-attention decoding heads to the main transformer backbone. In a single forward pass, these heads predict a sequence of candidate tokens in parallel. The main model then verifies these tokens, accepting a prefix of correct predictions and resampling from the first point of failure. The method is applied to the BPT model and demonstrates a ~1.7x speedup while preserving the original model's generation quality.

**Strengths:**

1. The paper successfully achieves its primary goal of speeding up inference for auto-regressive mesh generation. The reported 1.7x speedup is significant from a practical standpoint, and the empirical results convincingly show that this is achieved with a negligible drop in geometric quality compared to the base model.
2. The proposed multi-head architecture is a clever adaptation of speculative decoding that avoids the need to train and maintain a separate, smaller draft model. The technical execution, including the use of distillation, LoRA fine-tuning to align distributions, and a clear verification strategy, is sound and well-justified.
3. The authors provide a comprehensive set of ablations that analyze the impact of key design choices, such as the number of decoding heads, the type of decoder, and the acceptance criteria. This adds credibility to their results and provides valuable insights for future work in this specific direction.

**Weaknesses:**

1. The primary weakness of this work lies in its limited conceptual contribution. Speculative decoding is a well-established and widely known technique for accelerating inference in large language models. This paper presents a straightforward application of this existing "trick" to the domain of mesh generation. While the engineering adaptation is non-trivial, it does not introduce a new fundamental concept or paradigm. The contribution is more of an engineering speed-up than a core research advance, making it a questionable fit for a top-tier conference like ICLR that values foundational contributions.
2. The most critical and challenging research problem in generative 3D modeling today is the quality of the output—achieving correct topology, plausible geometric detail, global coherence, and semantic consistency. This paper explicitly sidesteps this primary challenge. By design, its quality ceiling is strictly limited by the performance of the base model (BPT). The paper even acknowledges that it inherits all the failure modes of BPT (Appendix A.9, A.10). Investing research effort into accelerating a potentially flawed or quality-limited generator seems to misalign with the community's most pressing needs.
3. While a 1.7x speedup is a welcome improvement, it is ultimately an incremental gain. It does not fundamentally alter the capabilities of auto-regressive mesh models or enable new applications that were previously out of reach. It makes a slow process slightly less slow, but does not represent a breakthrough that would significantly change the research landscape or practical workflows.

**Questions:**

1. The core contribution is applying a known LLM acceleration technique to meshes. Could the authors elaborate on what makes this adaptation uniquely challenging or insightful in the 3D domain, beyond the architectural changes? What fundamental principle does this work teach us about 3D generation itself, rather than just about inference optimization?
2. Given that generation quality is the main bottleneck for the field, how do the authors justify the importance of this work for ICLR? Why is accelerating a model of a certain quality level a more critical contribution at this stage than proposing a new method that could generate higher-quality meshes, even if slower?
3. The performance of XSpecMesh is capped by the base BPT model. Have the authors considered applying their method to a different, potentially higher-quality (even if experimental or slower) auto-regressive backbone? Is the technique general enough to apply to any such model, and would the speedup gains be consistent?
4. The introduction of D decoding heads and LoRA adds complexity and parameters during the training phase. Could you provide a more detailed analysis of the trade-offs, including training time, memory overhead, and implementation complexity, versus the achieved 1.7x inference speedup?

---

> ### Author Response · Authors · 2025-11-19
>
> We appreciate your positive and constructive feedback, including comments such as “practically meaningful, cleverly designed architecture, and thorough experimental validation.”
>
> **1. Challenges in the 3D domain (W1, Q1).**
>
> In LLMs, the typical acceleration strategy is to use a draft model to accelerate the base model. The draft model is designed to have far fewer parameters than the base model while maintaining a prediction distribution sufficiently close to that of the base model, so as to enable lossless acceleration. However, existing mesh generation models already have quite small parameter counts, making it difficult for an even smaller model to achieve accurate predictions. Therefore, designing a draft model in the mesh generation domain is challenging.
>
> Considering that mesh generation models have a much smaller vocabulary than LLMs, decoding over this vocabulary is relatively easier. This motivates us to design an even more lightweight model structure and adopt multi-head parallel prediction to achieve lossless acceleration.
>
> **2. Bottleneck in generation quality and limited speedup (W2, Q2, W3).**
>
> BPT and DeepMesh are currently SOTA open-source models. Our acceleration scheme is generic and can be well adapted as models continue to evolve, including future base models with larger parameter sizes. Therefore, we believe our work is both scientifically valuable and practically useful.
>
> **3. Generality of the method (Q3).**
>
> Our method is highly general and applies to various transformer-based autoregressive models. However, up to now, BPT remains the most stable open-source model in terms of generation quality. As mentioned in the paper, the open-sourced 0.5B version of DeepMesh has stability issues, and another higher-quality SOTA model, Mesh-RFT, has not yet been open-sourced.
>
> Since Mesh-RFT uses an uncompressed serialization scheme, it results in longer generation time and more redundant tokens. Therefore, we expect our method to achieve an even better speedup when applied to Mesh-RFT.
>
> To further demonstrate the generality of our method, we also apply it to DeepMesh, and the results are shown in the table below, where SCR denotes the step compression ratio.
> |              |   CD    |   HD    |  SCR  | Speedup|
> |--------------|---------|---------|-------|--------|
> | DeepMesh     | 0.1323  | 0.2648  | 1.000 |  1.00  |
> | DeepMesh+ours| 0.1378  | 0.2767  | 2.184 |  1.52  |
> | BPT          | 0.1165  | 0.2223  | 1.000 |  1.00  |
> | BPT+ours     | 0.1168  | 0.2261  | 2.021 |  1.71  |
>
> **4. Complexity introduced by multi-head decoding and LoRA fine-tuning (W4).**
>
> We provide the convergence time and memory usage during training for different numbers of decoding heads, as well as the memory usage and speedup during testing.
>
> (1) **Training: convergence time and memory usage for different numbers of decoding heads**
>
> | D (heads) | Convergence time     | Memory usage |
> |-----------|----------------------|--------------|
> | D = 2     | ~16 A800 hours       | 62672 MiB    |
> | D = 3     | ~16 A800 hours       | 64644 MiB    |
> | D = 4     | ~16 A800 hours       | 64688 MiB    |
> | D = 5     | ~16 A800 hours       | 66646 MiB    |
> | D = 6     | ~16 A800 hours       | 68604 MiB    |
>
> (2) **Testing: memory usage and speedup for different numbers of decoding heads**
>
> | Model     | Memory usage | Speedup |
> |-----------|--------------|---------|
> | BPT       | 1954 MiB     | 1.00    |
> | D = 2     | 1998 MiB     | 1.44    |
> | D = 3     | 2020 MiB     | 1.63    |
> | D = 4     | 2064 MiB     | 1.71    |
> | D = 5     | 2086 MiB     | 1.69    |
> | D = 6     | 2128 MiB     | 1.67    |
>
> (3) **Implementation complexity.**
>
> The implementation complexity is relatively low. We only need to add a multi-head sampling and verification module on top of the base model.

---

> ### Author Response · Authors · 2025-11-26
> **Kind Reminder to Review the Rebuttal**
>
> Dear Reviewer,
> Thank you very much for your insightful comments. I have provided detailed explanations and responses to the questions and weaknesses you mentioned. In addition, we have conducted supplementary experiments to further demonstrate the generality of our method and the value of its lossless acceleration. I would like to kindly remind you to take a look at the rebuttal when convenient. If you have any further concerns or would like to discuss anything in more detail, we would be very happy to continue the conversation.

---

> > ### Comment · Reviewer_HMRD · 2025-11-27
> > **Follow-up Questions on DeepMesh Implementation Details**
> >
> > Thank you for the response and the additional results.
> >
> > I have a specific technical question regarding the application to **DeepMesh**. Unlike BPT, DeepMesh employs an **Hourglass Transformer** architecture. Managing the **KV Cache** (specifically updates and rollbacks) during multi-head speculative decoding is non-trivial in this hierarchical structure.
> >
> > Could you elaborate on how the KV Cache is handled in this setting? I also maintain some reservations regarding the robustness of the fixed-threshold verification compared to standard sampling.
> >
> > Including these implementation details in the revised manuscript would greatly improve the reproducibility and clarity of your work.

---

> > > ### Author Response · Authors · 2025-11-28
> > >
> > > Thank you for your response.
> > >
> > > 1. **How to maintain the KV cache**
> > >
> > > Indeed, the hierarchical structure of the Hourglass Transformer introduces technical challenges for maintaining the KV cache.
> > >
> > > Specifically, in the Hourglass Transformer, the sequence is hierarchically downsampled and upsampled. We maintain a separate KV cache for each layer and record the *valid* KV cache positions (i.e., the positions of tokens that have been verified). Whenever we use the KV cache, we only access entries at valid positions. Once a token is verified, we update the valid positions in the KV cache in a layer-by-layer manner.
> > >
> > > It is worth noting that we do not adopt a rollback mechanism, but rather an overwrite mechanism. More precisely, we use hierarchical absolute positions to write into the KV cache. This means that at each forward pass we write the latest values into the corresponding KV cache positions, thereby overwriting any previously invalid entries. Here, invalid KV cache entries refer to tokens that have been rejected.
> > >
> > > Thank you again for this suggestion. We will update the supplementary material in the near future to include additional implementation details regarding the DeepMesh-based acceleration.
> > >
> > > 2. **Effect of the threshold on sampling and acceleration**
> > >
> > > Tokens whose scores exceed the threshold are verified and contribute to acceleration. For tokens below the threshold, we resample at that position, which effectively degenerates to standard sampling. As a result, the model’s sampling distribution remains very close to that of standard sampling.
> > >
> > > In the paper, we conduct an ablation study on fixed thresholds. The experiments show that a threshold of 0.5 achieves a good balance between acceleration and output quality. In Table 3, we also demonstrate that the robustness of samples generated by our method is comparable to that of the base model.

---

> > > > ### Comment · Reviewer_HMRD · 2025-11-28
> > > >
> > > > Thank you for providing these additional implementation details. The explanation of how the KV cache is managed and how the sampling threshold functions is very clear. I will raise my score to reflect this.

---

### Official Review · Reviewer_QJVU · 2025-10-31

**Soundness:** 3
**Presentation:** 3
**Contribution:** 2
**Rating:** 4
**Confidence:** 3

**Summary:**

The paper presents XSpecMesh, a framework designed to accelerate auto-regressive mesh generation models while maintaining the quality of the generated 3D meshes. It addresses the high inference latency of existing auto-regressive models, which require thousands of sequential next-token predictions. This is done by employing multiple cross-attention speculative decoding heads for multi-token prediction.

**Strengths:**

The core achievement is the reported 1.7x acceleration in generation speed without sacrificing output quality. The adaptation of using lora to reduce the performance gap is reasonable.

The paper is clear in defining the problem. The high latency of auto-regressive mesh generation, stemming from the thousands of required forward passes, is introduced as the primary motivation.

parallel prediction, verification/resampling, and LoRA integration are easy to follow.

The paper has a sensitivity analysis for the critical hyperparameters: the number of speculative heads ($D$) and the verification threshold ($\delta$).

**Weaknesses:**

While the method avoids truncated training, long sequences lead to massive KV cache usage. Instead, it pointed out that truncated training is a drawback because it loses the context. Then, the work could not achieve long token sequences that the truncated training is addressing. The mesh polycount might be limited.


The 1.7 times speed up actually might not be much.

**Questions:**

What is your maximum polycount?

Could you provide somewhere for testing the practical performance? thanks

---

> ### Author Response · Authors · 2025-11-19
>
> We appreciate your positive and constructive feedback, including comments such as “achieving speedup without loss of quality” and “the method is reasonable and effective.”
>
> **1. Avoiding truncated training and its impact on long-sequence support (W1).**
>
> This is an insightful question.
> (1) Using truncated training can indeed support longer sequence generation to some extent, but it may lose contextual information and lead to quality degradation [1].
>
> (2) On the other hand, since our method is a lossless acceleration approach, we need to keep the training setup as consistent as possible with the base model. The base model BPT does not use truncated training, so we follow the same training strategy. If the base model adopted truncated training, our method should also use truncated training to better align the distribution with the main model. We believe that whether the number of generated polygons is limited is primarily determined by the base model.
>
> (3) In addition, we have applied our method to DeepMesh, using the same truncated training strategy as DeepMesh. The resulting metrics are: CD: 0.1378, HD: 0.2767, SCR: 2.184, with an actual speedup of about 1.52×.
>
> |              |   CD    |   HD    |  SCR  | Speedup|
> |--------------|---------|---------|-------|--------|
> | DeepMesh     | 0.1323  | 0.2648  | 1.000 |  1.00  |
> | DeepMesh+ours| 0.1378  | 0.2767  | 2.184 |  1.52  |
> | BPT          | 0.1165  | 0.2223  | 1.000 |  1.00  |
> | BPT+ours     | 0.1168  | 0.2261  | 2.021 |  1.71  |
>
> **2. A 1.7× speedup may not be significant enough (W2).**
>
> Since our work focuses on lossless acceleration, a 1.7× speedup can still substantially improve generation efficiency. Compared with lossless acceleration methods for LLMs (around 2× speedup) [2,3], our speedup is mainly constrained by the following factors:
> (1) Compared with LLMs (7B, 13B, …), mesh autoregressive generation models have far fewer parameters (0.5B) and less parameter redundancy, which naturally limits the achievable speedup.
> (2) Mesh sequences exhibit strong dependencies. In mesh serialization, a single vertex is usually serialized into multiple tokens, whereas in LLMs one or more characters are often serialized into a single token. This leads to stronger dependencies among mesh tokens, which in turn constrains the speedup ratio.
> (3) BPT and DeepMesh adopt compressed sequence representations with low redundancy, which further restricts the achievable speedup.
>
> **3. Maximum number of supported polygons (Q1).**
>
> The maximum number of polygons that can be generated is determined by the base model. Our acceleration method can be applied to various mesh generation models. Taking BPT as an example, the maximum number of generated polygons is about 8K.
>
> **4. Links for performance testing (Q2).**
>
> Due to policy reasons, we are unable to provide external links at this time. We will release the code, including both inference and training code, after the paper is accepted.
>
> [1] Haohan Weng, Zibo Zhao, Biwen Lei, Xianghui Yang, Jian Liu, Zeqiang Lai, Zhuo Chen, Yuhong Liu, Jie Jiang, Chunchao Guo, et al. Scaling mesh generation via compressive tokenization. In Proceedings of the Computer Vision and Pattern Recognition Conference, pp. 11093–11103, 2025.
> [2] Charlie Chen, Sebastian Borgeaud, Geoffrey Irving, Jean-Baptiste Lespiau, Laurent Sifre, and John Jumper. Accelerating large language model decoding with speculative sampling. arXiv preprint arXiv:2302.01318, 2023.
> [3] Ziteng Sun, Ananda Theertha Suresh, Jae Hun Ro, Ahmad Beirami, Himanshu Jain, and Felix Yu. Spectr: Fast speculative decoding via optimal transport. Advances in Neural Information Processing Systems, 36:30222–30242, 2023.

---

> ### Author Response · Authors · 2025-11-26
> **Kind Reminder to Review the Rebuttal**
>
> Dear Reviewer,
>
> Thank you very much for your constructive comments. I have provided detailed explanations and responses to the questions and weaknesses you mentioned. I would like to kindly remind you to take a look at the rebuttal when convenient. If you have any further concerns or would like to discuss anything in more detail, we would be very happy to continue the conversation.

---

### Official Review · Reviewer_rZkX · 2025-10-31

**Soundness:** 3
**Presentation:** 3
**Contribution:** 2
**Rating:** 6
**Confidence:** 4

**Summary:**

This paper introduces a novel method, XSpecMesh, aimed at accelerating the inference process of auto-regressive 3D mesh generation models while preserving generation quality. Diverging from traditional methods that rely on single next-token prediction, XSpecMesh employs a multi-head speculative decoding scheme. The core idea is to utilize multiple lightweight decoding heads to predict several future tokens in parallel within a single forward pass. Furthermore, the authors have designed a verification and resampling strategy that leverages the backbone model to validate these predicted candidate tokens and resample any that do not meet the quality criteria. To improve the success rate of speculative predictions, the method incorporates a distillation strategy and LoRA fine-tuning to train the decoding heads, aligning their prediction distributions with that of the backbone model. Experimental results demonstrate that this method achieves approximately a 1.7× speedup over the baseline model (BPT) without sacrificing generation quality.

**Strengths:**

**1、Significant Speedup While Maintaining Quality:** The most significant contribution of this paper is the achievement of a ~1.7× inference speedup, which is a substantial improvement for time-consuming auto-regressive generation tasks. More importantly, this acceleration is accomplished with almost no loss in generation quality, as demonstrated by objective metrics like Chamfer Distance (CD) and Hausdorff Distance (HD), as well as the user study. This proves the method's effectiveness and practical value.

**2、Efficient and Novel Framework Design:** Unlike common speculative decoding schemes in Large Language Models (LLMs) that require a separate "draft model," the multi-head decoding approach proposed here is more lightweight and efficient. It avoids the complexity and overhead of training and maintaining an additional model. The clever integration of LoRA fine-tuning and knowledge distillation not only addresses the challenge of predicting multiple tokens directly from intermediate hidden states but also effectively aligns the decoding heads with the backbone model. This design is highly sophisticated.

**3、Thorough Experimental Validation:** The authors have conducted comprehensive experiments to validate their method's effectiveness. The ablation studies (Table 3), in particular, are well-designed and clearly demonstrate the contributions of key components, such as the cross-attention decoders and LoRA fine-tuning, to the final performance (in terms of both quality and speedup). Additionally, the analysis of hyperparameters like the number of decoding heads (D) and the acceptance threshold (δ) provides valuable insights for the application and optimization of the method.

**Weaknesses:**

**1、Limited Acceleration Factor:** Although a 1.7× speedup is valuable in the context of mesh generation, this figure appears conservative when compared to speculative decoding techniques in the LLM domain, where speedups of 3-4× or even higher are common. The paper would benefit from a deeper discussion on the bottlenecks that prevent a higher acceleration factor in this task. For instance, is it due to stronger dependencies in mesh sequences, lower entropy in the token distribution, or the computational overhead introduced by the multi-head architecture?

**2、Limited Novelty:** The core ideas of the paper (multi-head prediction, verification, distillation) are heavily inspired by mature techniques from the LLM acceleration field. While successfully applying these concepts to 3D mesh generation is a valuable contribution, from an algorithmic perspective, the work appears more like a successful "A+B" application rather than a fundamentally new theoretical framework. The paper needs to better articulate the unique challenges of the mesh generation task and why these challenges necessitate specific adaptations to existing speculative decoding frameworks to highlight its originality.

**3、Performance Ceiling is Limited by the Base Model:** The proposed method is essentially an "accelerator," and its upper bound for generation quality is entirely determined by the chosen base model (BPT in this case). As shown in the appendix (Figure 10, Visualization of failure cases), XSpecMesh inherits all the generation deficiencies of the BPT model. This means the method can make a good model faster, but it cannot fix or improve the model's inherent flaws. This limitation should be more explicitly discussed in the main body of the paper.

**Questions:**

1、According to Figure 5(a), the speedup plateaus and even slightly decreases when the number of decoding heads (D) exceeds 4. What do the authors believe is the main bottleneck causing this phenomenon? Is it that the acceptance rate of candidate tokens has reached its limit, or does the computational overhead from the additional decoding heads begin to outweigh the benefits of parallel prediction?

2、The statement of a 1.7× speedup in this paper lacks sufficient rigor. It is recommended to include the average time required to generate 100 face to more accurately demonstrate the claimed acceleration.

3、LoRA fine-tuning plays a crucial role in your method. Could the authors provide more analysis or intuition on how LoRA specifically helps the model predict multiple future tokens? For example, does LoRA primarily alter the information representation of the Transformer's final hidden state (hs) to better encapsulate sequence information for multiple future tokens?

4、Table 1 indicates that the vocabulary size for mesh generation models is considerably smaller than for language models. How does this characteristic impact the design of speculative decoding? Does it make a "multi-head" approach more feasible than a "draft model" approach? Conversely, does this small vocabulary introduce other unique challenges for improving the token acceptance rate?

---

> ### Author Response · Authors · 2025-11-19
>
> We appreciate your positive and constructive feedback, including “maintaining quality while significantly accelerating,” “efficient and novel architecture design,” and “thorough experimental validation.”
>
> **1. Limited speedup factor (W1).**
>
> Thank you very much for the insightful question. It can be addressed from three perspectives:
> (1) Compared with LLMs (7B, 13B, …), our mesh autoregressive generation model has a much smaller number of parameters (0.5B) and less parameter redundancy, which naturally limits the achievable speedup.
> (2) As you mentioned, mesh sequences exhibit strong dependencies. In mesh serialization, a single vertex is usually serialized into multiple tokens, whereas in LLMs one or more characters are often serialized into a single token. This leads to stronger dependencies among mesh tokens, which in turn constrains the speedup ratio.
> (3) BPT and DeepMesh adopt compressed sequence representations with low redundancy, which further restricts the achievable speedup. On the other hand, the overhead brought by multiple decoding heads is relatively small (about 2 ms per decoding head), so this is not the main bottleneck.
>
> **2. Limited novelty (W2).**
>
> A draft model is typically designed such that its parameter size is much smaller than that of the base model, while its predicted distribution is sufficiently close to the base model so as to provide lossless acceleration. However, existing mesh generation models are already quite small in terms of parameter count. Designing an even smaller model that still achieves accurate prediction is difficult. Therefore, it is challenging to design an effective draft model in the mesh generation setting.
> Considering that the vocabulary size of mesh generation models is much smaller than that of LLMs, decoding over this smaller vocabulary is easier. This motivates us to design an even more lightweight model structure and employ multi-head parallel prediction to achieve lossless acceleration.
>
> **3. Performance limited by the base model (W3).**
>
> Thank you very much for this suggestion. We acknowledge this point. Since our method is a lossless acceleration approach, it inevitably inherits the failure cases of the base model. We have discussed this limitation in the supplementary material, and we will move this part into the main text in the next revision.
>
> **4. Too many decoding heads lead to a drop in speedup (Q1).**
>
> The main bottleneck is that tokens at farther positions are harder to predict and have lower acceptance rates. As a result, the overall acceptance rate of candidate tokens has reached a ceiling; additional decoding heads only introduce extra computation overhead, which reduces the effective speedup.
>
> **5. Average time to generate 100 faces (Q2).**
>
> On average, BPT takes 8.75 seconds to generate 100 faces, ours takes 4.86 seconds, and DeepMesh takes 5.47 seconds. When measured by the number of generated faces, our speedup ratio is 1.8×. We will add an extra column in Table 2 of the main paper to present this metric.
> |                | Time per 100 Faces (s) |
> |----------------|------------------------|
> | BPT            |         8.75           |
> | ours           |         4.86           |
> | DeepMesh       |         5.47           |
>
> **6. LoRA fine-tuning for better prediction of multiple future tokens (Q3).**
>
> LoRA helps the model predict multiple future tokens from two aspects:
> (1) As you pointed out, in the standard autoregressive paradigm, the transformer is trained to predict the next token, and causal masking hides future tokens. Thus, the hidden state \(h_s\) mainly encodes information for the immediate next token, making it difficult to directly decode multiple future tokens from the original \(h_s\). LoRA fine-tuning alters the latent features in \(h_s\), enabling the additional decoding heads to better decode multiple future tokens from it.
> (2) LoRA fine-tuning better aligns the distributions of the backbone model and the decoding heads, making the backbone more inclined to accept the tokens predicted by the decoding heads.
>
> **7. Small vocabulary in mesh generation models (Q4).**
>
> This property allows us to use a simpler structure for speculative decoding. Yes, we believe that the multi-head scheme is more feasible than a draft model, mainly for two reasons:
> (1) A draft model must be sufficiently small to provide acceleration. However, the current base models already have relatively small parameter sizes, so an even smaller draft model struggles to achieve the required prediction accuracy.
> (2) A draft model is typically an independent transformer, whose training is both difficult and computationally expensive. Given the relatively small vocabulary size, our lightweight multi-head design is a more practical and effective choice.

---

> ### Author Response · Authors · 2025-11-26
> **Kind Reminder to Review the Rebuttal**
>
> Dear Reviewer,
> Thank you sincerely for your positive evaluation and constructive comments. I have provided detailed explanations and responses to the questions and weaknesses you mentioned. I would like to kindly remind you to take a look at the rebuttal when convenient.
> Thank you again for your time.

---

> > ### Comment · Reviewer_rZkX · 2025-11-26
> >
> > Thank you very much for the author's reply. The author has addressed all my questions. I believe this paper deserves acceptance, so I have raised my score.

---

> > > ### Author Response · Authors · 2025-11-27
> > >
> > > We sincerely appreciate you raising the rating. Your recognition has greatly encouraged us. Thank you again!

---

### Author Response · Authors · 2025-11-30

**General Response**

Dear Area Chair, we would like to provide a brief summary of the rebuttal process below.

All three reviewers agree that:

1. The method achieves a **1.7× lossless speedup** without sacrificing generation quality.
2. The **multi-head speculative decoding architecture**, **LoRA-based fine-tuning**, and the **systematic ablation and hyperparameter analysis** are key strengths of the paper.

The main concerns focus on the **type of contribution** and the **magnitude of the speedup**. In the rebuttal, we:

1. **Systematically analyzed** the differences between mesh generation and LLMs in terms of parameter scale, vocabulary size, sequence redundancy, and dependency structure.
2. **Presented acceleration results** on both BPT and DeepMesh, together with detailed speed and memory usage metrics.
3. **Clarified the generality** of our approach: as base models become more powerful, our method can still provide quality-preserving acceleration.

After the rebuttal:

1. **Reviewer rZkX and Reviewer HMRD both explicitly raised their scores** (with confidence levels 4 and 5 respectively, indicating strong familiarity with the area).
   - Reviewer rZkX explicitly supports acceptance and raised the score from **6 to 8**.
   - Reviewer HMRD stated that they were raising their score and are now **inclined toward acceptance**, acknowledging that the technical details are sufficient and the method is clearly described.

2. Reviewer QJVU, after the additional experiments, did not raise any new questions or objections.

---

### Note · Authors · 2026-01-26

I have read and agree with the venue's withdrawal policy on behalf of myself and my co-authors.

---

### Meta-Review · Area_Chair_bYcf · 2026-01-06

**Summary:**

The initial ratings for this paper are 1x marginal accept and 2x marginal reject. The reviewers mentioned that the strengths are: 1) the proposed approach achieved 1.7x speed up while maintaining the performance. 2)  The proposed multi-head architecture is a clever adaptation of speculative decoding that avoids the need to train and maintain a separate, smaller draft model. 3) Extensive experiments and ablations to validate the proposed method and designs. The major weaknesses pointed out by the reviewers (both reviewers who were negative initially) are 1.7x speed up is not significant, limited novelty and too many decoding heads. Although the last reviewers agreed to raise the scores further in the discussion phase, it is not clear that the rebuttal and discussion adequately address the doubt on the significance of 1.7x speedup and the novelty, as the discussion was steered towards KV cache and sampling threshold functions instead. The second reviewer did not take part in the discussion and remains negative. Considering that the 1.7x speedup issue is not clearly resolved and it is the core part of the paper, the AC decides to reject the paper.

**Reviewer Concerns:**

The major weaknesses pointed out by the reviewers are 1.7x speed up is not significant, limited novelty and too many decoding heads are addressed by the rebuttal. Further questions asked by the last reviewer about KV cache and fixed threshold are also resolved during the discussion. The doubt on 1.7x speed up is not significant and novelty remain outstanding and not fully clarified and resolved in the rebuttal and discussion.

**Reviewer Scores:**

The first reviewer would raise to 8, the second reviewer would maintain 4, and the last reviewer would raise to 6 or 8.

---

### Decision · Program_Chairs · 2026-01-26

Reject